# Efficient and Stable Air-Processed Ternary Organic Solar Cells Incorporating Gallium-Porphyrin as an Electron Cascade Material

**DOI:** 10.3390/nano13202800

**Published:** 2023-10-21

**Authors:** Anastasia Soultati, Maria Verouti, Ermioni Polydorou, Konstantina-Kalliopi Armadorou, Zoi Georgiopoulou, Leonidas C. Palilis, Ioannis Karatasios, Vassilis Kilikoglou, Alexander Chroneos, Athanassios G. Coutsolelos, Panagiotis Argitis, Maria Vasilopoulou

**Affiliations:** 1Institute of Nanoscience and Nanotechnology (INN), National Center for Scientific Research Demokritos, Agia Paraskevi, 15341 Athens, Greece; a.soultati@inn.demokritos.gr (A.S.); marwver@gmail.com (M.V.); e.polydorou@inn.demokritos.gr (E.P.); karmadorou@gmail.com (K.-K.A.); zoegeorgiopoulou@gmail.com (Z.G.); i.karatasios@inn.demokritos.gr (I.K.); v.kilikoglou@inn.demokritos.gr (V.K.); 2Department of Physics, University of Patras, 26504 Rio Patra, Greece; lpalilis@upatras.gr; 3Solid State Physics Section, Physics Department, National and Kapodistrian University of Athens, Panepistimioupolis, 15784 Athens, Greece; 4Department of Electrical and Computer Engineering, University of Thessaly, 38221 Volos, Greece; alexander.chroneos@imperial.ac.uk; 5Department of Materials, Imperial College, London SW7 2AZ, UK; 6Laboratory of Bioinorganic Chemistry, Department of Chemistry, University of Crete, Voutes Campus, P.O. Box 2208, 71003 Heraklion, Greece

**Keywords:** gallium porphyrin, electron cascade, ternary organic solar cell, exciton dissociation efficiency, photostability

## Abstract

Two gallium porphyrins, a tetraphenyl GaCl porphyrin, termed as (TPP)GaCl, and an octaethylporphyrin GaCl porphyrin, termed as (OEP)GaCl, were synthesized to use as an electron cascade in ternary organic bulk heterojunction films. A perfect matching of both gallium porphyrins’ energy levels with that of poly(3-hexylthiophene-2,5-diyl) (P3HT) or poly[N-9′-heptadecanyl-2,7-carbazole-alt-5,5-(4′,7′-di-2-thienyl-2′,1′,3′-benzothiadiazole)] (PCDTBT) polymer donor and the 6,6-phenyl C_71_ butyric acid methyl ester (PCBM) fullerene acceptor, forming an efficient cascade system that could facilitate electron transfer between donor and acceptor, was demonstrated. Therefore, ternary organic solar cells (OSCs) using the two porphyrins in various concentrations were fabricated where a performance enhancement was obtained. In particular, (TPP)GaCl-based ternary OSCs of low concentration (1:0.05 vv%) exhibited a ~17% increase in the power conversion efficiency (PCE) compared with the binary device due to improved exciton dissociation, electron transport and reduced recombination. On the other hand, ternary OSCs with a high concentration of (TPP)GaCl (1:0.1 vv%) and (OEP)GaCl (1:0.05 and 1:0.1 vv%) showed the poorest efficiencies due to very rough nanomorphology and suppressed crystallinity of ternary films when the GaCl porphyrin was introduced to the blend, as revealed from X-ray diffraction (XRD) and atomic force microscopy (AFM). The best performing devices also exhibited improved photostability when exposed to sunlight illumination for a period of 8 h than the binary OSCs, attributed to the suppressed photodegradation of the ternary (TPP)GaCl 1:0.05-based photoactive film.

## 1. Introduction

Today’s energy crisis has caused a shock of unprecedented breadth and intricacy, leading to the evaluation of renewable energy opportunities that meet the continuous electricity demand. Hence, exploiting solar energy for power generation has become even more imperative nowadays, shaping the photovoltaic research landscape [1,2,3,4,5,6]. Given the significance of briskly diversifying our energy supply and enhancing energy efficiency, overcoming the hurdles concerning global organic solar cell (OSC) promotion should be a priority for all interested parties involved, from academics and publishers to companies and funding agencies. Among different photovoltaic technologies, OSCs are thin-filmed solar cells with unique features for either outdoor [7,8,9] or indoor use [10,11,12,13,14]. The possibility of their application for large area coverage [15,16,17,18], their semitransparent character [19,20,21], and their low-cost fabrication by printing methods [18,22,23,24], enable OSCs to be integrated into buildings’ facades and windows or sunroofs. In this way, OSCs can provide energy derived from sunlight that might either warm a room or contribute to electric power (i.e., hybrid vehicles). [25,26,27,28]. Furthermore, due to their lightweight nature [29,30], flexibility [31,32] and coloration [33], OSCs can be an affordable solution for delivering smaller quantities of electricity required for wearable [34,35,36] and portable electronics [37]. On the other hand, indoor OSCs appear to have great application potential in the Internet of Things [38,39] and energy harvesting [40,41]. 

OSCs could also play a crucial role in the reduction of electronic waste, known as e-waste, which is rapidly accelerating in our modern way of life [42]. The annual production of e-waste is globally estimated to be over 50 million tons; therefore, green and sustainable technologies are urgent. Not only low-toxic organic materials and solvents, but also recycling substrates have been used for the fabrication of OSCs [43,44]. Zhou et al. [45] demonstrated the first efficient and recyclable organic solar cell on cellulose nanocrystal substrates. The separation of the fabricated OSCs into their major components with low-energy and low-temperature processing paved the way for a fully recyclable, sustainable and eco-friendly solar cell technology. 

Undoubtedly, OSCs provide the momentum for a large number of technological applications. In recent years, sustained research efforts have notably increased the solar-to-electrical power conversion efficiencies of OSCs to a satisfying level for them to be brought to market. The most efficient semitransparent OSC yielded a power conversion efficiency (PCE) of 19% [46], while a record high efficiency of 28% for indoor OSC has also been reported [47]. Nevertheless, despite the fact that OSCs have reached the performance of their predecessors, unfortunately their commercialization is restricted by stability issues. Instability in OSCs is dictated by a series of degradation modes that are usually assigned to the stability of individual materials or interfaces [48,49,50,51]. For instance, many organic compounds are chemically degraded when constantly irradiated, mainly owing to their susceptibility to atmospheric oxygen [52] and moisture [53]. Additionally, morphological degradation may occur under extreme environmental conditions, such as extended light exposure and elevated temperatures. Climate variability that meets outdoor operation can lead to the diffusion of buffer layers and electrodes, and especially to the evolution of the phase-segregated bulk-heterojunction (BHJ) active layer due to the strong mobility of organic materials, stressing the cell and eventually causing its decaying performance [48,54].

It is well-known that ideal active layer morphology, aside from having a favorable molecular orientation to optimize charge transport and being compositionally structured in vertical direction to facilitate charge collection, should simultaneously form a bicontinuous network of mixed nanodomains of semiconducting materials for efficient exciton dissociation and suppression of charge recombination [55,56,57,58,59]. Accordingly, the morphology of active layers plays a pivotal role in impacting the overall photovoltaic performance. While many strategies, including material optimization [60,61], device engineering [62] and encapsulation processes [63], have been proposed to improve and stabilize active layer morphology, structural engineering is identified as a strategic priority. In this context, numerous reports have emerged that focus on designing effective additives to enhance the electronic and/or morphological properties of typical OSC systems with the commonly used poly(3-hexylthiophene) (P3HT) and phenyl-C_61_-butyric acid methyl ester (PC_61_BM) as donor and acceptor, respectively [64,65,66,67]. 

The approach of adding a third component to form ternary BHJ blends for solar cells entails some specific intermolecular interactions in terms of controlling morphology. The third component should also act as a cascade layer reducing the charge transfer energy offsets at the door/acceptor lowest unoccupied molecular orbital (LUMO) levels, resulting in reduced recombination losses. In addition, the cascade material should have excellent electron acceptor and transport properties for the case of the formation of a heterojunction between the third component and the donor polymer. In this vein, porphyrins appear to be suitable candidates as components that could be functional BHJ additives. Owing to their strong absorption properties (high extinction coefficients) in both the blue (~400–550 nm, Soret or B-band) and red (~700–900 nm, Q-band) regions of the visible spectrum, their high thermal stability and their remarkable electron transfer capability, porphyrins have been successfully applied in organic and planar perovskite solar cells as cathode interfacial modifiers, realizing efficient device operation and longevity [68,69]. Following this thinking, porphyrins could be introduced as a cascade energy material in a binary BHJ blend inducing efficient charge extraction as they can provide more efficient charge transport pathways [70]. Herein, we demonstrate the effect of two gallium porphyrins, tetraphenyl GaCl porphyrin, termed as (TPP)GaCl, and an octaethylporphyrin GaCl porphyrin, termed as (OEP)GaCl, in various ratios, as additives in two different binary photoactive blends. The organic blends consist of poly(3-hexylthiophene-2,5-diyl) (P3HT) or poly[N-9′-heptadecanyl-2,7-carbazole-alt-5,5-(4′,7′-di-2-thienyl-2′,1′,3′-benzothiadiazole)] (PCDTBT) polymer donors with the 6,6-phenyl C_71_ butyric acid methyl ester (PCBM) fullerene acceptor. In this study, the photophysical properties of the target porphyrins, alone and within the binary host BHJ blends, were studied by means of optical spectroscopy, while Fourier-transform infrared (FTIR) spectroscopy was used to analyze the chemical composition in every case and the possible interaction between the three components of the ternary films. Additionally, electrochemical characterization of the porphyrin compounds was performed to estimate the highest occupied molecular orbital (HOMO) and lowest unoccupied molecular orbital (LUMO) levels of the two porphyrins, revealing the potential role of GaCl porphyrins as electron cascade materials. Therefore, ternary organic solar cells using both porphyrins as the third component inserted in the PCDTBT:PCBM or P3HT:PCBM binary blend in various concentrations were fabricated. Note that all ternary and binary OSCs were prepared in air (without using glove box), at room temperature (25°) and under 40% humidity (without using glove box) to promote a low-cost fabrication procedure. The addition of (TPP)GaCl with low concentrations (1:0.05 vv%, see Section 4 for more details) increased the PCE of the ternary OSCs by ~17% in comparison with the efficiency of the binary devices, which were assigned to the enhanced electron transport and exciton dissociation into free charges, along with the reduced recombination. On the other hand, when (TPP)GaCl with higher concentrations or (OEP)GaCl was added to the ternary film, a rougher morphology surface and suppressed crystallinity were observed leading to poor PCE values of the corresponding ternary OSCs. The ternary (TPP)GaCl 1:0.05 vv%-based OSC also exhibited enhanced photostability under exposure to sunlight illumination for a period of 8 h, attributed to the better structural stability of the ternary films than the binary layer. 

## 2. Results and Discussion

### 2.1. Characterization of (TPP)GaCl and (OEP)GaCl Porphyrin Materials

Two gallium porphyrin materials, a tetraphenyl GaCl porphyrin, termed as (TPP)GaCl, and an octaethylporphyrin GaCl porphyrin, termed as (OEP)GaCl, were synthesized following a previous procedure [71] to use as the third component in ternary organic BHJ films. In brief, to a solution of porphyrin (1.9 mmol) and GaCl (9 mmol) in 100 mL of acetic acid, sodium acetate (25 mmol) was added in 200 mL of acetic acid. The mixture was refluxed for 24 h, and, after returning to ambient temperature, the solid precipitate was collected, dried and recrystallized in a toluene–heptane mixture. Prior to the incorporation of the porphyrins as a third component in the binary blends P3HT:PCBM and PCDTBT:PCBM, their optical and electrochemical properties were investigated. It should be mentioned that these molecules were selected due to their fine solubility in the same organic solvent of the organic blends [72]. The molecular structure of the gallium porphyrins, which constitute four phenyl (C_6_H_5_) groups attached to the pyrrole rings [71], are shown in Figure 1a, and their Fourier-transform infrared and UV-Vis absorption spectra are displayed in Figure 1b,c, respectively. 

In the FTIR transmittance spectrum of the (TPP)GaCl, weak peaks at around 3050 cm^−1^ corresponding to the C-H stretching vibrations of aromatic groups are observed. Moreover, peaks at 1107 cm^−1^ and 802 cm^−1^ are attributed to the C-H scissoring of phenyl rings in and out of the plane bending mode of the pyrrole rings, respectively, while peaks at 701 cm^−1^ and 754 cm^−1^ are assigned to the out-of-plane bending of the phenyl rings. In the area from 1347 cm^−1^ to 1600 cm^−1^, multiple peaks corresponding to the stretching vibration of the C=C aromatic groups of phenyl are observed, while in the 1175–1206 cm^−1^ region the stretching mode of C=N in pyrrole rings appears. The peak of the Ga-Cl bond is not observed in the spectrum, since it can be found in the region 330–370 cm^−1^ [73], which is out of the range of our FTIR instrument. On the other hand, in the FTIR spectrum of (OEP)GaCl, the C=C stretching mode of the phenyl rings and the C-N stretching mode of the pyrrole groups are detected at the 1300–1580 cm^−1^ region and at 1150 cm^−1^, respectively. The bands ascribed to the substitution of the pyrrole unit correspond to the vibration modes of the ethyl group. Specifically, those peaks are found at 2966 cm^−1^, 2930 cm^−1^ and 2869 cm^−1^ (C-H stretching), in the 960–1160 cm^−1^ region (C-H bending), at 842 cm^−1^ (C-H out-of-plane bending) and at 749 cm^−1^ (rocking mode of CH_2_ group). 

Figure 1c shows the UV-Vis absorption spectra of both porphyrins spin-coated on a glass from a chloroform solution with a concentration of 1 mg mL^−1^. It is observed that both gallium porphyrins exhibit typical Soret bands in the range of 375–475 nm, and the Q-bands are all located at 515–625 nm revealing further electronic transitions in the molecular orbitals. In particular, the (TPP)GaCl porphyrin exhibits a sharper Soret band centered at 438 nm and two weak Q bands at 555 and 595 nm, while the (OEP)GaCl porphyrin has a broader Soret band located at 413 nm and two Q bands at 541 and 577 nm.

In order to examine the possible use of these porphyrin compounds as electron cascades in BHJ blends, their energy levels (E_HOMO_ and E_LUMO_) were estimated by performing cyclic voltammetry. Figure 2a–d shows the cyclic voltammograms of (TPP)GaCl and (OEP)GaCl porphyrins deposited on indium tin oxide (ITO)/glass substrates. Oxidation and reduction potentials are summarized in Table 1 together with the estimation of the corresponding HOMO and LUMO levels evaluated by using the following empirical formulas [74]:(1)$$E_{HOMO}=-(E_{|\overset{}{}onset,ox\overset{}{}vs\overset{}{}F_{c}^{+}/F_{c}|}+5.1)\;(\mathrm{eV})$$
(2)$$E_{LUMO}=-(E_{|\overset{}{}onset,red\overset{}{}vs\overset{}{}F_{c}^{+}/F_{c}|}+5.1)\;(\mathrm{eV})$$ where $E_{|\overset{}{}onset,ox\overset{}{}vs\overset{}{}F_{c}^{+}/F_{c}|}$ and $E_{|\overset{}{}onset,red\overset{}{}vs\overset{}{}F_{c}^{+}/F_{c}|}$ are the oxidation and reduction potential onsets in respect to ferrocene, respectively, defined as the position where the current starts to differ from the baseline. From the cyclic voltammograms the $E_{|\overset{}{}onset,ox\overset{}{}vs\overset{}{}F_{c}^{+}/F_{c}|}$ value is +0.5 V and +0.4 V for (TTP)GaCl and (OEP)GaCl, respectively, which results in HOMO levels of −5.6 eV for (TTP)GaCl and −5.5 eV for (OEP)GaCl. The LUMO levels of (TTP)GaCl and (OEP)GaCl are −3.8 eV and −3.9 eV, respectively, and are determined by the reduction potential onset (−1.3 V for (TTP)GaCl and −1.2 V for (OEP)GaCl). The energy diagram illustrated in Figure 2e reveals the perfect matching of both gallium porphyrins’ energy levels with that of the polymer donors (PCDTBT or P3HT) and the PCBM acceptor, forming an efficient cascade system that could facilitate electron transfer between donor and acceptor. 

### 2.2. Characterization of Ternary Blend Films 

Figure 3a,b and Appendix A presents the UV-Vis absorption spectra of ternary PCDTBT:PCBM:Ga-porphyrin and P3HT:PCBM:Ga-porphyrin films, respectively, forming on a glass substrate from a chloroform solution, as described in the Section 4. The binary PCDTBT:PCBM and P3HT:PCBM are also shown for comparison reasons. UV-Vis absorption spectra of the PCDTBT:PCBM film display a peak centered at 381 nm and a broad absorption band from 500 nm to 640 nm (Figure 3a,b), while the P3HT:PCBM spectrum exhibits three absorption regions of P3HT at 501 nm, 545 nm and 597 nm, and extends into the visible region from approximately 400 nm to 700 nm (Appendix A). In the spectra of the ternary films, the peak corresponding to the Soret band of both porphyrin compounds is clearly seen. Moreover, the blue shift of the PCDTBT:PCBM and P3HT:PCBM spectra is observed when (TTP)GaCl and (OEP)GaCl porphyrins are incorporated into binary blends, which is more pronounced in the case of the (OEP)GaCl-based ternary films, and especially in the P3HT:PCBM:GaCl.

In order to investigate the chemical structure of the prepared ternary films, FTIR transmittance measurements were performed. In the FTIR spectra of PCDTBT:PCBM:(TPP):GaCl, shown in Figure 3c, a combination of peaks attributed to all three components of the ternary film are observed. In particular, the peaks at 2925 cm^−1^, 2852 cm^−1^ (C-H stretch), at 1739 cm^−1^ (C=O stretch), at 1494 cm^−1^ (C=C stretch), in the region 1330–1430 cm^−1^ (CH_2_ and CH_3_ bend), at 1250 cm^−1^ (C-O stretch) and at 1106 cm^−1^ (C-O bend) are assigned to the binary PCDTBT:PCBM blend. The only apparent (TPP)GaCl absorption band in the ternary film is detected in the aromatic stretching region, which can be seen in the spectrum of the sample with a 1:0.1 vv% ratio. In addition, the pyrrole breathing oscillation peak is detected at 1007 cm^−1^. In addition, some weak peaks under 1000 cm^−1^, corresponding to the porphyrins, are apparent (802, 754 and 701 cm^−1^). On the other hand, the porphyrin peaks located at 3050 cm^−1^ in the 1175–1206 cm^−1^ region and in the 1107–1110 cm^−1^ region cannot be observed in the ternary film, due to the low concentration of the (TPP)GaCl porphyrin with respect to the binary blend. Similarly, in the case of PCDTBT:PCBM:(OEP)GaCl (Figure 3d), the C-H stretching peaks just below 3000 cm^−1^, located at 2966, 2925 and 2852 cm^−1^, are a combination of the peaks from the binary blend and the porphyrin additive corresponding to the same vibration mode. The carbonyl peak at 1739 cm^−1^ remains unchanged, while the C=C stretching peaks of the porphyrin overlap with all other polymer blend peaks found in the region, more specifically at 1494 cm^−1^ (C=C stretch), 1430–1330 cm^−1^ (CH_2_ and CH_3_ bend) and 1250 cm^−1^ (C-O stretch). Just below this region, the C-O bending peak at 1106 cm^−1^ can be seen, while the C-N stretching peak of the porphyrin is absent from the ternary film. Furthermore, porphyrin weak peaks at 1160–960 cm^−1^ (C-H bend ethyl) are depicted. The same results can be also concluded for the ternary P3HT:PCBM:(TPP)GaCl and P3HT:PCBM:(OEP)GaCl ternary films, the FTIR transmittance spectra of which are shown in Appendix A, respectively. 

Moreover, photoluminescence (PL) measurements were applied to probe the efficiency of the exciton dissociation and charge transfer. Figure 3e,f and Appendix A illustrate the photoluminescence steady-state emission (PL) spectra of various samples, including pristine films and blends of PCDTBT and P3HT with (TPP)GaCl and (OEP)GaCl in ratios of 1:0.05 and 1:0.1. In the PL spectrum of the pure P3HT film, a peak corresponding to excitonic emission is observed at 653 nm (Appendix A). Upon introducing (TPP)GaCl to the P3HT blend, the PL intensity decreases, indicating that efficient charge transfer occurred from P3HT and (TPP)GaCl. On the other hand, the addition of (OEP)GaCl results in increased fluorescence of P3HT, potentially attributable to the aggregation of (OEP)GaCl molecules. The well-known tendency of porphyrin molecules to aggregate could form plasmonic nanostructures in the ternary blends. Plasmonic effects through metal nanostructures enhance photon absorption and charge transfer leading to an improved photocurrent due to hot electron generation in the metal nanostructure [75,76]. Plasmon formation using GaCl-based porphyrin molecules is currently being investigated and the corresponding results will be demonstrated in a subsequent work. Additionally, a shift towards longer wavelengths (redshift) is noted in the PL spectrum of P3HT, particularly at higher concentrations of (OEP)GaCl. This shift may be ascribed to the presence of trapped states within the energy gap of (OEP)GaCl, aligning with the lowest energy states in the highest occupied molecular orbital (HOMO) of P3HT. Consequently, notable recombination occurs between photoinduced holes originating from the lowest HOMO states of P3HT and deep traps of (OEP)GaCl, along with enhanced photoinduced electron back transfer from the LUMO to the HOMO of P3HT.

Figure 3e shows the PL spectra of the PCDTBT and PCDTBT:(TPP)GaCl films demonstrating a significant decrease in emitted light intensity as the porphyrin concentration increases. Adding (TPP)GaCl in a ratio of 1:0.05 vv% reduces the fluorescence intensity of the PCDTBT film by 30%, and with a ratio of 1:0.1 vv% the fluorescence is almost completely suppressed by 88%. The most pronounced quenching occurs with (OEP)GaCl, likely due to the aggregation of its molecules, which serves as an additional factor in reducing the fluorescence. Photoluminescence quenching indicates efficient energy transfer from the polymer PCDTBT to the porphyrin compound. However, despite the quenching effect, a shift towards shorter wavelengths (blue shift) is observed in the PCDTBT emission spectrum after incorporating the porphyrins. This shift could be attributed to the mitigation of charge-carrier-trapping states within the porphyrin, resulting in a modified emission for PCDTBT. PL shift also indicates that the trap-assisted recombination mechanism is dominated in the ternary photoactive layers. Furthermore, time-resolved PL (TRPL) measurements, presented in Figure 3f for the PCDTBT:(TPP)GaCl and PCDTBT:(OEP)GaCl, show a shorter exciton life time, suggesting the faster exciton dissociation, especially in the case of the PCDTBT:(TPP)GaCl 1:0.1 and PCDTBT:(OEP)GaCl 1:0.05 samples. In the case of the P3HT blended with the two GaCl-porphyrins (Appendix A), a short exciton lifetime, facilitating exciton dissociation, exhibits only the P3HT:(TPP)GaCl film. 

In order to study the surface nanomorphology of the ternary films, atomic force microscopy (AFM) measurements were performed. Figure 4 and Appendix A depict the 5 × 5 μm^2^ surface topographic AFM image of pristine PCDTBT:PCBM and P3HT:PCBM films, respectively, as well as the films of these photoactive blends modified with (TPP)GaCl and (OEP)GaCl in ratios of 1:0.05 and 1:0.1. AFM analysis of the PCDTBT:PCBM film (Figure 4a) revealed that the introduction of (TPP)GaCl in both concentrations (1:0.05 and 1:0.1) slightly changed the surface roughness of the samples exhibiting root mean square (RMS) roughness of 1.63 nm and 1.00 nm for the PCDTBT:PCBM:(TPP)GaCl 1:0.5 (Figure 4b) and PCDTBT:PCBM:(TPP)GaCl 1:0.1 (Figure 4c), respectively, compared with the RMS = 0.99 of the binary film. However, small grains of a few nanometers, which may be assigned to the porphyrin compound, are observed on the surface of the ternary films. On the other hand, the nanomorphology of the ternary films based on the (OEP)GaCl porphyrin is significantly changed. The surface of the PCDTBT:PCBM:(OEP)GaCl (Figure 4d,e) consists of large domains, with size ~500 nm for the PCDTBT:PCBM:(OEP)GaCl 1:0.1 and RMS = 14.94 nm. A similar trend in the surface nanomorphology of the porphyrin-modified P3HT:PCBM film is also observed (Appendix A). The pristine P3HT:PCBM film exhibited a smooth surface with RMS = 0.99 nm, while the incorporation of (TPP)GaCl resulted in a slightly rough surface with small grains. However, the addition of (OEP)GaCl in the binary P3HT:PCBM resulted in phase separation of the materials consisting of the ternary films along with large domains and rougher surfaces, with RMS of 8.62 nm and 20.09 nm for the P3HT:PCBM:(OEP)GaCl 1:0.05 and P3HT:PCBM:(OEP)GaCl 1:0.1 films, respectively. On the other hand, the incorporation of gallium-porphyrins in the binary P3HT:PCBM blend significantly affected the crystallinity of the ternary films. Appendix A shows the XRD patterns of binary and P3HT:PCBM-based ternary films. It is observed that P3HT:PCBM:(TPP)GaCl 1:0.05 exhibit improved crystallinity with a larger crystalline size of 7.9 nm (estimated with the Scherrer equation) compared with 6.1 nm for the binary blend. However, a higher concentration of (TPP)GaCl (1:0.1) in the binary film suppresses the crystallization of the ternary film. Crystallinity suppression of the P3HT:PCBM-based ternary films was also observed upon incorporation of the (OEP)GaCl in the binary blend, which may affect charge transport in the ternary film. On the other hand, due to the amorphous nature of the PCDTBT polymer, it was not possible to extract similar results from the corresponding X-ray diffraction diagrams (Figure 4f), where no changes in the crystallinity of the PCDTBT:PCBM are observed when (TPP)GaCl or (OEP)GaCl introduced in the binary blend. 

In order to investigate the compatibility of the two porphyrins with the donor and acceptor materials, the surface energies of the neat (TPP)GaCl, (OEP)GaCl, PCDTBT, P3HT and PCBM films were measured. The surface energy of material X (γ_X_) was estimated using contact angle measurements. The contact angle of deionized and diiodomethane droplets on the porphyrins PCDTB, P3HT and PCBM’s films are shown in Appendix A, and the surface energy of each material is summarized in Appendix A. The surface energy of (TPP)GaCl (~41 mN m^−1^) is closer to that of the donor and acceptor materials (~40.2, 45.7, and 49.5 mN m^−1^, for the P3HT, PCDTBT and PCBM, respectively) than that of the (OEP)GaCl suggesting that the porphyrin (TPP)GaCl has good miscibility with the BHJ materials [77,78]. On the other hand, (OEP)GaCl film exhibits a high surface energy (68.1 mN m^−1^) leading to phase separation and the formation of large aggregates. The location of the porphyrins in the binary blended films P3HT:porphyrin, PCDTBT:porphyrin and PCBM:porphyrin were also investigated. Appendix A show the contact angle measurements of the binary films P3HT:porphyrin, PCDTBT:porphyrin and PCBM:porphyrin, and Appendix A shows the surface energy of the binary films versus the concentrations of (TPP)GaCl and (OEP)GaCl, respectively. It is observed that the surface energy of binary films based on a (TPP)GaCl porphyrin compound is similar to those of neat PCDTBT, P3HT and PCBM films. Therefore, we next estimated the wetting coefficient (ω) for the ternary films to predict the location of the porphyrins in them using Young’s equation [79]. Appendix A present the contact angle measurements of the P3HT:PCBM and PCDTBT:PCBM based ternary films with (TPP)GaCl and (OEP)GaCl as the third component. As shown in Appendix A, the ω in the all ternary blends (PCDTBT:PCBM:(TPP)GaCl, PCDTBT:PCBM:(OEP)GaCl, P3HT:PCBM:(TPP)GaCl and P3HT:PCBM:(OEP)GaCl) is between −1 and 1 (−1 < ω < 1), demonstrating that the porphyrin compound is located at the donor/acceptor interface [80],which could improve charge transfer in the ternary OSCs. 

Another goal, apart from the optoelectronic, morphological and structural investigation of GaCl porphrins-based ternary films, was the stability study of the prepared films under constant illumination conditions. Therefore, binary and ternary films were exposed to sunlight illumination in ambient conditions, and UV-Vis and FTIR spectroscopy, along with XRD measurements, were performed. To assess the degree to which porphyrins contribute to the long-term stability of each photoactive film under ambient conditions, it was necessary to conduct an investigation into the samples when traces of oxygen or humidity were present. Figure 5 shows the UV-Vis absorption spectra of pristine PCDTBT:PCBM and porphyrin-modified blend films, where no differences before and after their exposure to 8 h of sunlight illumination are noticed, indicating no degradation of the binary and ternary films. On the other hand, in the case of P3HT:PCBM:GaCl (Appendix A) a blue shift of ~10 nm of the three main peaks of P3HT after exposure to sunlight illumination for 8 h can be seen. Note that the peak corresponding to the Soret band of the GaCl porphyrins remains unaffected.

Appendix A show the FTIR transmittance spectra of fresh and exposed to sunlight illumination for 8 h of the P3HT:PCBM:GaCl and PCDTBT:PCBM:GaCl films, respectively. According to Appendix A, the peak at 820 cm^−1^ for the P3HT:PCBM photoactive film is slightly shifted to higher energies when the film is exposed to sunlight illumination (8 h), suggesting a suppression of charge transfer from sulfur atoms of P3HT to PCBM. Moreover, the P3HT:PCBM:(TPP)GaCl and P3HT:PCBM:(OEP)GaCl films remain stable upon sunlight illumination as no significant changes in the FTIR spectra are observed (Appendix A). In the case of PCDTBT:PCBM-based binary and ternary films, FTIR analysis of all spectra (Appendix A) reveals that there are no noticeable changes even after 8 h of light exposure. 

More interestingly, the P3HT:PCBM:(TPP)GaCl film exhibited better structural stability upon porphyrin compared with the P3HT:PCBM film. Figure 6a,b shows the XRD patterns of the fresh and exposed P3HT:PCBM and P3HT:PCBM:(TPP)GaCl films, respectively. In the case of the binary film, the diffraction peak at 2θ = 5.16° attributed to the (100) plane of P3HT is shifted to lower angle values (2θ = 4.7°) resulting in increased d-spacing (from 1.71 nm to 1.88 nm) and a reduced crystalline size of ~2.4 nm, thus degradation of the exposed P3HT:PCBM film to sunlight illumination. On the other hand, P3HT:PCBM:(TPP)GaCl 1:0.05 is more resistant to sunlight, since a small change in its crystallinity is observed. In particular, the d-spacing is slightly increased from 1.71 nm to 1.73 nm (estimated by Bragg law, where 2θ = 5.1°), while the crystalline size is reduced to ~5.5 nm (crystalline size of fresh film was 7.9 nm), indicating improved photostability. 

### 2.3. Fabrication of Ternary-Based Organic Solar Cells

According to these results, organic solar cells (OSCs) with ternary photoactive layers based on (TPP)GaCl porphyrin of low concentration (1:0.05) were prepared. The device structure shown in Figure 7a along with the chemical structure of the polymer donors and fullerene acceptor constituting the active layer was ITO/PEDOT:PSS/P3HT:PCBM:(TPP)GaCl 1:0.05/Al or ITO/PEDOT:PSS/PCDTBT:PCBM(TPP)GaCl 1:0.05/Al. Note that OSCs with binary photoactive layers (without the addition of the (TPP)GaCl porphyrin) were also fabricated for comparison reasons. Figure 7b,c depict the current density–voltage characteristic curves of binary and ternary OSCs based on the PCDTBT:PCBM and P3HT:PCBM blends, respectively. Enhanced device performance of about ~17% is observed for the ternary devices with the (TPP)GaCl porphyrin as the third component of the active layer, compared to the binary OSCs. In particular, the device with the ternary PCDTBT:PCBM:(TPP)GaCl 1:0.05 active layer exhibited higher short-circuit current density (J_SC_) and fill factor (FF) values of 11.14 mA cm^−2^ and 0.53, respectively, and thus lead to a higher power conversion efficiency (PCE) value of 4.61%, compared with the corresponding binary (PCDTBT:PCBM) OSC with J_sc_ = 10.66 mA cm^−2^, FF = 0.48, and PCE = 3.94% (Table 2). It is noted that this PCE enhancement is independent from the ternary film thickness since the insertion of porphyrin of low concentration in the binary blend did not change it. In the same context, improved J_sc_ of 12.90 mA cm^−2^, FF of 0.54, and PCE of 4.25% are observed for the P3HT:PCBM:(TPP)GaCl 1:0.05 device, while the binary OSC without the porphyrin compound exhibited lower values (J_SC_ = 12.09 mA cm^−2^, FF = 0.50, and PCE = 3.63%). These results were highly reproducible as shown in Appendix A, where the statistical errors and the standard box plots of the J_SC_, V_OC_, FF and PCE values resulting from the measurements in a batch of 12 identical devices for each kind are presented. The J_SC_ and FF enhancement suggests reduced recombination losses when the (TPP)GaCl prophyrin with low concentration is inserted in the active layer, which is also supported by the PL measurements (Figure 3e,f and Appendix A), as well as the higher shunt resistance (R_SH_) and the lower series resistance (R_S_) of the ternary OSCs compared with the binary devices, also summarized in Table 2. PL quenching indicates charge transfer between the polymer-donor (PCDTBT or P3HT) and the porphyrin (TPP)GaCl, which is also supported by the increased FF [81]. In addition, it is known that R_SH_ is related to charge recombination; therefore, higher R_SH_ values in the ternary OSCs reflects the suppression of trap-assisted recombination resulting in the enhancement of FF, and thus in the ternary OSCs PCE value. Note that an insignificant improvement in the V_OC_ of the ternary devices is also observed and assigned to the slight influence of the (TPP)GaCl 1:0.05 on the nanomorphology and crystallinity of the ternary films, as revealed in the AFM and XRD measurements. On the other hand, when the concentration of the (TPP)GaCl increased (1:0.01), a lower performance of the ternary OSCs was obtained, as shown in Appendix A. Poor electrical parameters were also observed for the ternary devices based on the (OEP)GaCl porphyrin (Appendix A), which can be attributed to the rougher surface morphology and the suppressed crystallinity of the ternary active layers (Figure 4 and Appendix A). The formation of large aggregates in the ternary films were observed when porphyrin compounds with high concentration were inserted in the binary photoactive films, which hindered exciton generation and charge separation, thus leading to lower PCE values. 

The reduced recombination of charge carriers in the ternary photoactive films compared with the binary layers was also confirmed by measuring the J-V characteristic curves of the OSCs under dark condition, which are shown in Appendix A. It is observed that the dark current density (J_DARK_) of the ternary OSCs at the reverse bias is reduced in respect to the binary device suggesting the suppression of the charge carriers in the ternary layer since the J_DARK_ is an indication of charge carrier recombination. Furthermore, the ternary devices exhibit higher J_DARK_ at forward bias compared with the binary OSCs leading to decreased R_S_. Lower R_S_ values favor charge extraction in the ternary OSCs. Additionally, the ideality factor of the ternary devices is improved (especially in the case of PCDTBT:PCBM-based OSCs), suggesting reduced charge carrier recombination. 

To further support that the (TPP)GaCl porphyrin acts as an electron cascade material when inserted in the binary PCDTBT:PCBM and/or P3HT:PCBM blends leading to improved charge transport, external quantum efficiency (EQE) measurements were performed. The EQE spectra of the ternary OSCs based on the PCDTBT:PCBM:(TPP)GaCl 1:0.05 and P3HT:PCBM:(TPP)GaCl 1:0.05 active layer along with the EQE spectra of the corresponding binary devices are shown in Figure 7d,e, respectively. It is observed that although the (TPP)GaCl porphyrin contributes to the absorption of the ternary films (Figure 3a and Appendix A), the light harvesting of the ternary OSCs remained unaffected by the insertion of the (TPP)GaCl in the BHJ blend (PCDTBT or P3HT). The shape of EQE spectra of the binary and ternary devices exhibited no changes; however, the ternary OSCs showed improved EQE compared with the binary devices, which could be attributed to the formation of an efficient cascade system in the ternary layers. Moreover, the J_SC_ of all OSCs estimated by the EQE measurements are 10.42 mA cm^−2^ and 11.05 mA cm^−2^ for the PCDTBT:PCBM and PCDTBT:PCBM:(TPP)GaCl devices, respectively, while for the P3HT:PCBM based OSCs the J_SC_ for the binary and ternary devices was 11.61 mA cm^−2^ and 12.25 mA cm^−2^, respectively. This small difference (less than 5%) between the measured and estimated J_SC_ suggest the very good accuracy of the fabricated OSCs electrical characterization measurements. 

In order to investigate the influence of (TPP)GaCl porphyrin on the charge generation efficiency, photocurrent density (J_PH_) versus effective voltage (V_EFF_) were plotted. J_ph_ is estimated from J_PH_ = J_ILL_ − J_DARK_, where J_ILL_ and J_DARK_ are the current density values under illumination and in dark condition, respectively. V_EFF_ is defined as V_EFF_ = V_0_ − V, where V_0_ is the voltage at which the J_PH_ is zero and V is the applied voltage. The estimated results of the binary and ternary OSCs based on the PCDTBT:PCBM and P3HT:PCBM blends are shown in Figure 7f,g, respectively. It is observed that the ternary OSCs with the (TPP)GaCl porphyrin exhibit higher J_PH_ values at low V_EFF_ compared with the corresponding binary devices, suggesting improved exciton dissociation into free carriers. In addition, both ternary OSCs show higher charge dissociation probabilities (P(E,T)–V_EFF_ presented in Appendix A) and charge collection efficiency (estimated from J_PH_/J_PH,SAT_, where J_PH,SAT_ is the saturated J_PH_) at short-circuit and at the maximum power point, respectively, than the binary devices. Moreover, at high V_EFF_ values (V_EFF_ > 1), the J_PH,SAT_ of the ternary OSCs is higher than those of the binary ones, resulting in an enhanced exciton generation rate (G_MAX_, estimated by the G_MAX_ = J_PH,SAT_/q L, where q is the elementary charge and L is the thickness of the photoactive layer) when (TPP)GaCl porphyrin is inserted in the binary blends. The higher exciton dissociation and charge extraction probabilities are consistent with the increase in FF in the ternary OSCs. 

Other than the device efficiency, the photostability of the (TPP)GaCl-based ternary OSCs was also investigated by exposing the fabricated binary and ternary OSCs to sunlight illumination for 8 h without any encapsulation. Figure 8 shows the variation of normalized PCE values over a period of 8 h illumination for the binary and ternary OSCs. It is observed that the PCDTBT:PCBM:(TPP)GaCl 1:0.05-based OSC retained over 85% of its initial PCE value, while the efficiency of the binary device decreased to ~71% after exposure to sunlight for 8 h (Figure 8a). Similar results were also obtained for the ternary P3HT:PCBM-based OSC, where the P3HT:PCBM:(TPP)GaCl 1:0.05 device exhibited enhanced photostability (retain the ~86% of its initial PCE value) compared with the exposed binary P3HT:PCBM OSC with its PCE decreasing to ~73% of its initial efficiency value (Figure 8b). This improved photostability can be attributed to the suppressed degradation, better nanomorphology and crystallinity of the ternary films after being exposed to sunlight illumination for 8 h. 

## 3. Conclusions

In conclusion, ternary air-processed photoactive layers consisting of a polymer donor (PCDTBT or P3HT) and a fullerene acceptor (PCBM) have been prepared using two GaCl-based porphyrins as the third component. Electrochemical measurements revealed a good energy level alignment between both gallium porphyrins, polymer donors (PCDTBT or P3HT) and the PCBM acceptor forming an efficient cascade system that could facilitate electron transfer between donor and acceptor. Therefore, ternary OSCs were fabricated. It was shown that the devices with the (TPP)GaCl 1:0.05 (i.e., PCDTBT:PCBM:(TPP)GaCl 1:0.05 or P3HT:PCBM:(TPP)GaCl 1:0.05) exhibited increased J_SC_ and FF, and thus higher PCE compared to the corresponding binary OSCs, assigned to the reduced trap-assisted recombination and improved exciton dissociation. On the other hand, the devices with higher (TPP)GaCl concentration (1:0.1), as well as the OSCs based on (OEP)GaCl, exhibited low performance due to the significant alteration of the nanormophology of the ternary films with very rough surfaces and the suppressed crystalline formation. The devices with the (TPP)GaCl 1:0.05-ternary layer also exhibited enhanced photostability when the OSC was exposed to sunlight illumination, maintaining the ~85% of the initial PCE value after 8 h, which is attributed to the better crystallinity and suppressed photodegradation of the ternary photoactive film. The proposed ternary approach presents a simple low-cost solution-processed strategy under ambient conditions to improve OSC efficiency which could also be compatible with recyclable substrates for sustainable, scalable and environmentally friendly energy production.

## 4. Experimental Section

**Film preparation.** Binary P3HT:PCBM (1:0.8 wt% ratio) and PCDTBT:PCBM (1:4 wt% ratio) films were prepared by spin-coating the binary blended solutions from 10 mg mL^−1^ of chloroform under air conditions (40% humidity at 25 °C). For the ternary films, 5 mg of each porphyrin molecule were dissolved in 1 mL of chloroform. Then, binary:porphyrin 1:0.05 vv% ratio and 1:0.1 vv% ratio solutions were prepared forming the P3HT:PCBM:(TPP)GaCl, P3HT:PCBM:(OEP)GaCl, PCDTBT:PCBM:(TPP)GaCl and PCDTBT:PCBM:(OEP)GaCl films through spin-coating. In particular, PCDTBT:PCBM-based binary and ternary films were formed through spin-coating at 1000 rpm for 2 min followed by thermal annealing at 60 °C for 1 min. In the case of P3HT:PCBM and P3HT:PCBM:GaCl porphyrin, spin-coating was performed at 600 rpm for 40 s, and then the samples were place on a hot plate at 135 °C for 10 min. The thickness of the P3HT:PCBM and PCDBTB:PCBM-based films was 100 ± 8 nm and 150 ± 5 nm, respectively. Note that all films were prepared under ambient conditions (40% humidity at 25 °C). 

**Thin film characterization methods.** The thickness of the binary and ternary films was measured with a profilometer. Fourier-transform infrared (FTIR) spectroscopy was performed using a Bruker Tensor 27 spectrometer equipped with a DTGS detector. Absorption spectra of the porphyrin, binary and ternary films were taken with a Perkin Elmer Lambda 40 UV-Vis spectrometer. Cyclic voltammetry of the two porphyrin molecules deposited on glass/ITO substrates were obtained with a VersaSTAT 4 potentiometer operating at a scan rate of 0.1 V s^−1^; the solvent was acetonitrile containing 0.1M LiClO_4_ as the electrolyte, while Ag/AgCl was the reference electrode. Steady-state photoluminescence (PL) spectra were recorded with a commercial platform (ARKEO—Cicci Research); the substrate was illuminated with a diode-pumped solid-state Nd:YVO4 + KTP laser (peak wavelength 532 nm ± 1 nm, with optical power of 1 mW on a circular spot of 2 mm in diameter: 31 mW cm^−2^) at an inclination of 45°. The fluorescence on the opposite side of the substrate was focused on a bundle of fibers (10 mm in diameter) with an aspheric lens close to the substrate to maximize the PL. The bundle sends the signal to a CCD-based spectrometer. Integration time and the number of averaging was maintained the same to better compare the results. Time-resolved PL (TRPL) spectra were measured with an FS5 spectrofluorometer from Edinburgh Instruments using a 478.4 nm laser as an excitation source. The surface morphology of the binary P3HT:PCBM and PCDTBT:PCBM and ternary P3HT:PCBM:porphyrin and PCDTBT:PCBM:porphyrin films was investigated with atomic force microscopy (AFM) using an NT-MDT AFM system in tapping operation mode. X-ray diffractograms (XRD) were recorded using a Siemens D500 diffractometer with Cu Kα radiation. 

**OSCs fabrication and characterization.** Indium tin oxide (ITO, coated on glass substrates, were cleaned by sonication in a deionized water, acetone and isopropyl bath for 10 min each. Then, they were subjected to UV-ozone for 20 min. Next, a PEDOT:PSS (poly(3,4-ethylenedioxythiophene)−poly(styrenesulfonate)) solution filtered with a 0.45 μm PVDF (polyvinylidene fluoride) filter was spin-coated on the ITO at 4000 rpm for 45 s, and the prepared films were placed on a hotplate at 150 °C for 30 min. Air-processed PCDTBT:PCBM and P3HT:PCBM with and without the GaCl porphyrins were spin-coated on PEDOT:PSS to prepare the ternary and binary organic solar cells. The devices completed with the deposition of aluminum served as the cathode electrode. The current density–voltage (J–V) characteristic curves of the all, binary and ternary OSCs were measured with a Keithley 2400 source-meter in the dark and under illumination with intensity of 100 mW cm^−2^ using a Xe lamp as the illumination source equipped with a AM 1.5G filter in ambient conditions. To accurately define the active area of all devices, we used aperture masks during the measurements with their area equal to those of the Al contacts (12.56 mm^2^).

## Figures and Tables

**Figure 1 nanomaterials-13-02800-f001:**
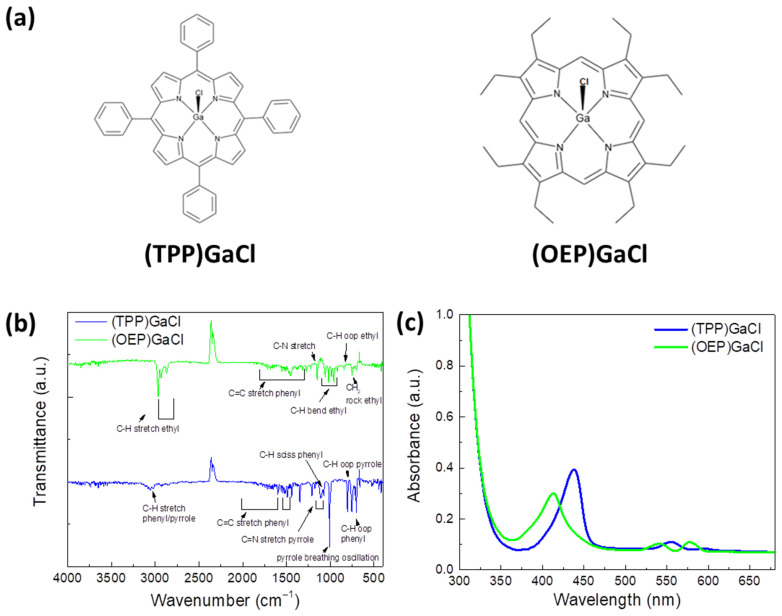
(**a**) Molecular structure of (TPP)GaCl and (OEP)GaCl porphyrins. (**b**) FTIR transmittance and (**c**) UV-Vis absorption spectra of the corresponding porphyrin thin films.

**Figure 2 nanomaterials-13-02800-f002:**
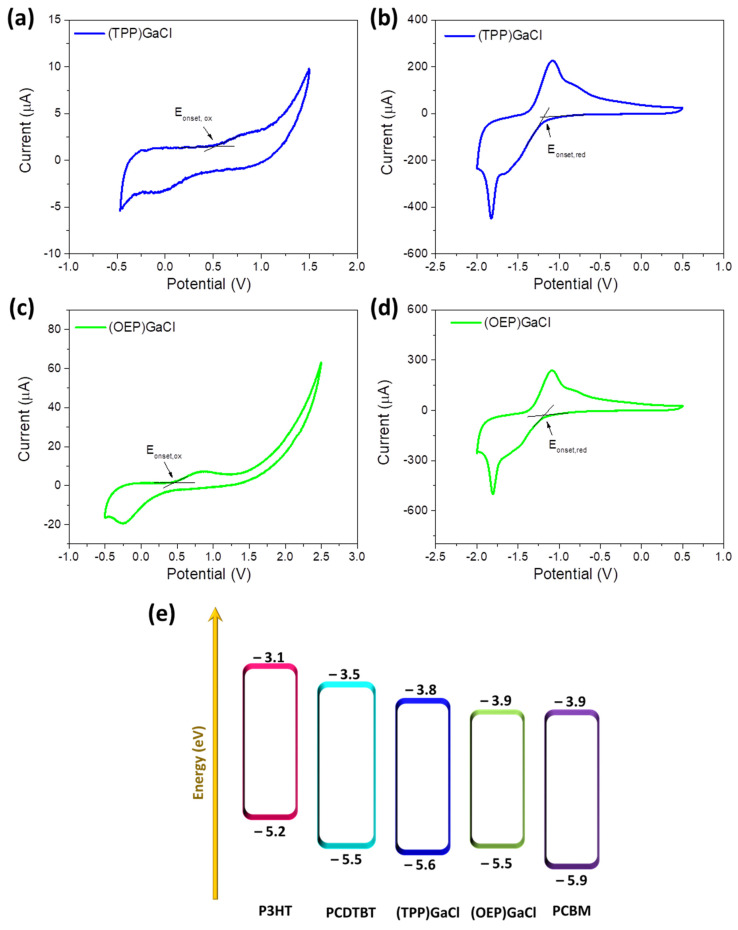
Cyclic voltammograms of (**a**,**b**) (TPP)GaCl and (**c**,**d**) (OEP)GaCl porphyrins. (**a**,**c**) refers to the oxidation processes, while (**b**,**d**) to the reduction processes. (**e**) Schematic energy level diagram of the organic semiconductors and gallium–porphyrin compounds illustrating the functionalization of porphyrins as electron cascade.

**Figure 3 nanomaterials-13-02800-f003:**
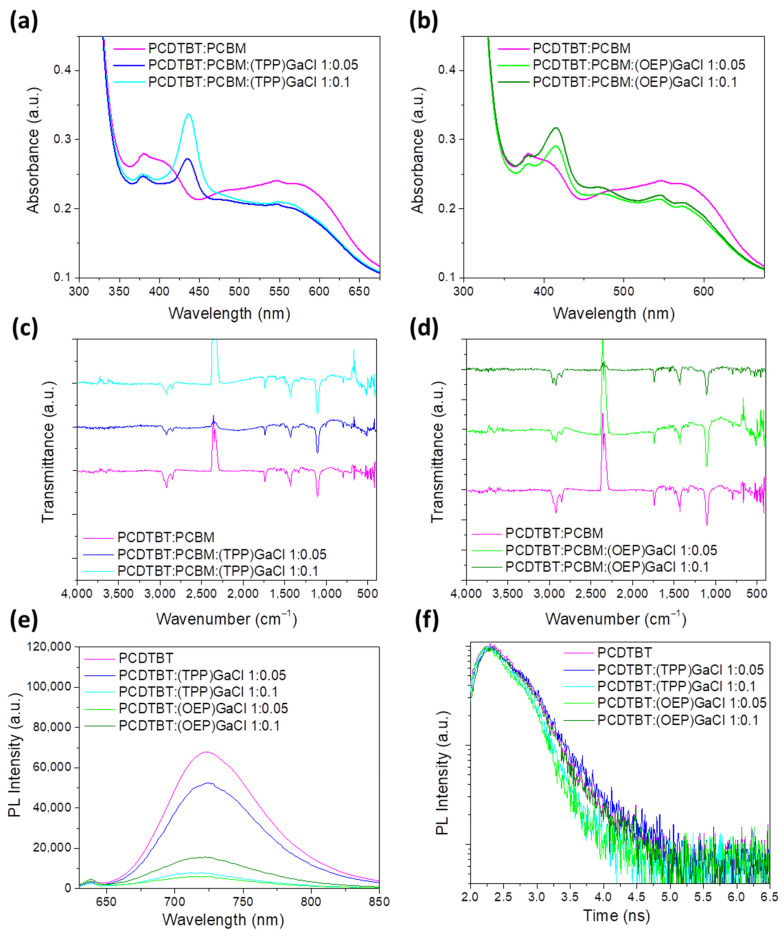
UV-Vis absorption spectra of binary and ternary PCDTBT:PCBM films with porphyrin additives: (**a**) (TPP)GaCl and (**b**) (OEP)GaCl in various ratios. FTIR spectra of binary and ternary PCDTBT:PCBM films with porphyrin additives: (**c**) (TPP)GaCl and (**d**) (OEP)GaCl in various ratios. (**e**) Steady-state PL spectra of PCDTBT without or with (TPP)GaCl and (OEP)GaCl in various ratios and (**f**) transient PL dynamics of the same samples.

**Figure 4 nanomaterials-13-02800-f004:**
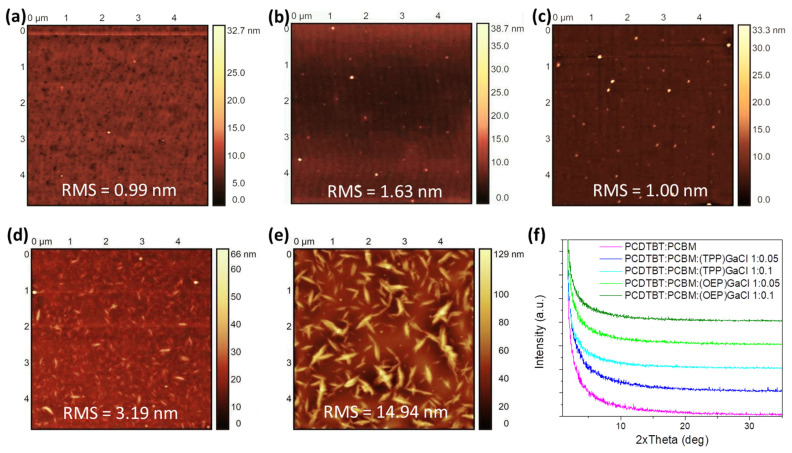
5 × 5 μm^2^ AFM height images of (**a**) binary PCDTBT:PCBM, (**b**) PCDTBT:PCBM:(TPP)GaCl 1:0.05 (**c**) PCDTBT:PCBM:(TPP)GaCl 1:0.1, (**d**) PCDTBT:PCBM:(OEP)GaCl 1:0.05 and (**e**) PCDTBT:PCBM:(OEP)GaCl 1:0.1 ternary blended films. (**f**) XRD patterns of the same samples.

**Figure 5 nanomaterials-13-02800-f005:**
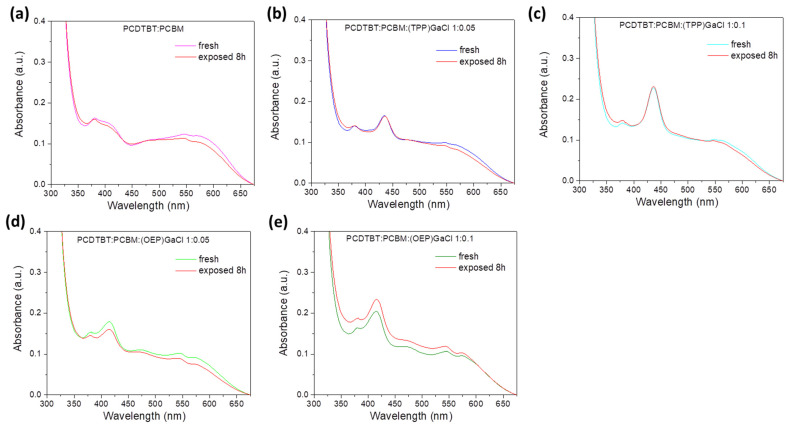
UV-Vis absorption spectra of fresh and exposed to sunlight illumination for 8 h of (**a**) binary PCDTBT:PCBM and ternary (**b**) PCDTBT:PCBM:(TTP)GaCl 1:0.05, (**c**) PCDTBT:PCBM:(TTP)GaCl 1:0.1, (**d**) PCDTBT:PCBM:(OEP)GaCl 1:0.05, and (**e**) PCDTBT:PCBM:(OEP)GaCl 1:0.1 films.

**Figure 6 nanomaterials-13-02800-f006:**
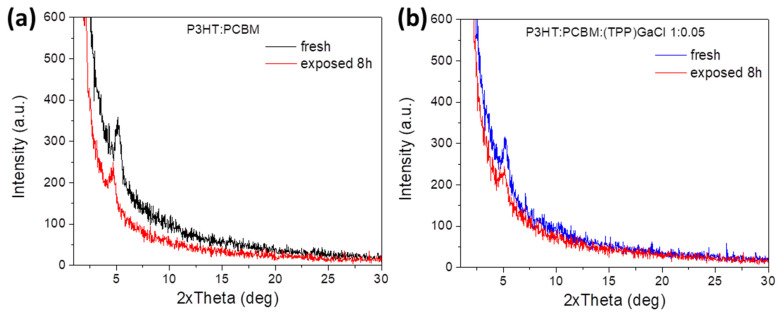
XRD patterns of fresh and exposed to sunlight illumination for 8 h of (**a**) binary P3HT:PCBM and ternary (**b**) P3HT:PCBM:(TPP)GaCl 1:0.05 films.

**Figure 7 nanomaterials-13-02800-f007:**
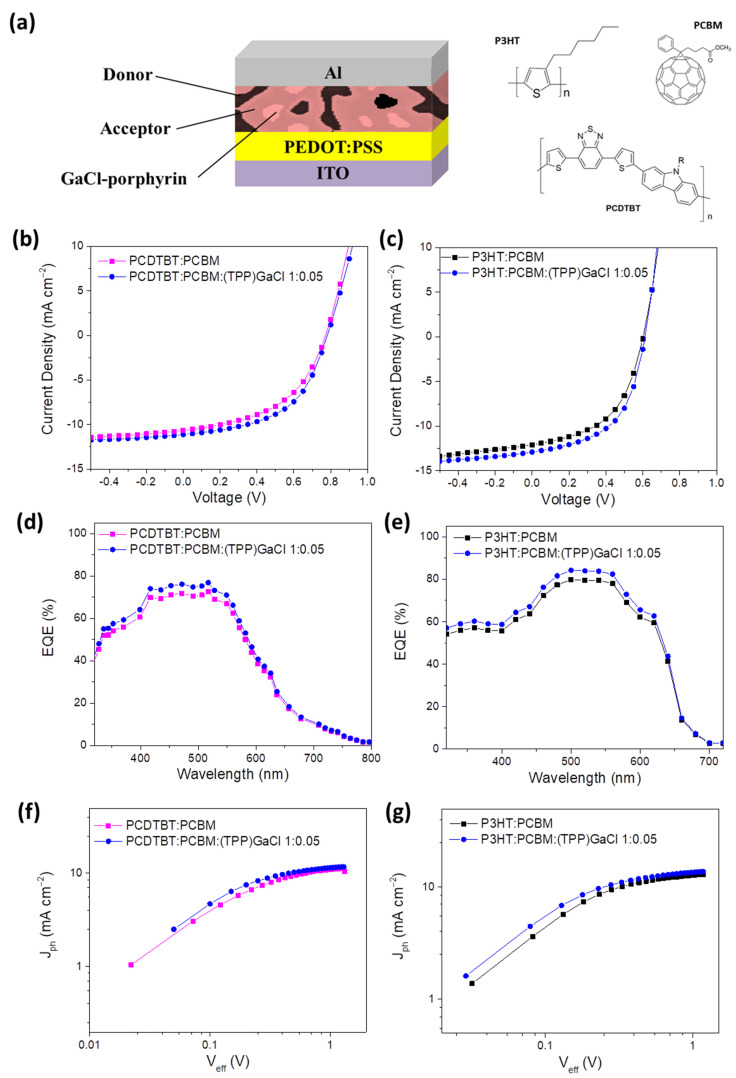
(**a**) Device structure and chemical structure of the donor and acceptor organic materials used in the photoactive layer of the ternary OSCs. J-V characteristic curves under 1.5 AM illumination of ternary OSCs based on (**b**) PCDTBT:PCBM:(TPP)GaCl 1:0.05 and (**c**) P3HT:PCBM:(TPP)GaCl 1:0.05 layers. J-V curves of binary devices are also depicted. (**d**,**e**) EQE spectra of the same devices. (**f**,**g**) Variation of photocurrent density (J_ph_) with effective voltage (V_eff_) of the same devices.

**Figure 8 nanomaterials-13-02800-f008:**
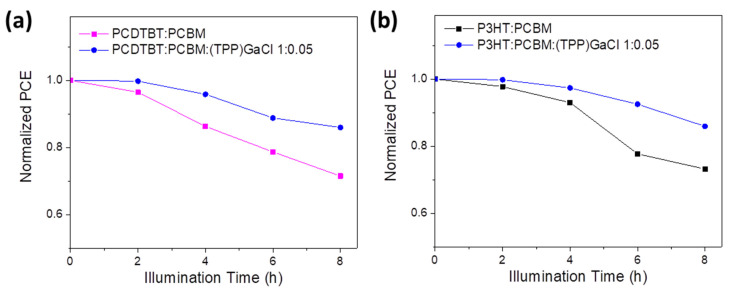
Stability study of binary and ternary OSCs. Normalized PCE values versus illumination time of OSCs based on (**a**) PCDTBT:PCBM:(TPP)GaCl 1:0.05 and (**b**) P3HT:PCBM:(TPP)GaCl 1:0.05 layers, compared with the corresponding binary devices.

**Table 1 nanomaterials-13-02800-t001:** Cyclic voltammetry calculated data of porphyrins.

Porphyrin	E_ox_ (eV)	E_red_ (eV)	E_HOMO_	E_LUMO_
**(TTP)GaCl**	0.5	−1.3	−5.6	−3.8
**(OEP)GaCl**	0.4	−1.2	−5.5	−3.9

**Table 2 nanomaterials-13-02800-t002:** Electrical parameters of the fabricated OSCs estimated by the J-V characteristic curves shown in Figure 7b,c (mean values and standard deviations were extracted from a batch of 12 independent devices).

Active Layer	J_SC_ (mA cm^−2^)	V_OC_ (V)	FF	PCE_aver._ (%)	R_S_ (Ω cm^2^)	R_SH_ (Ω cm^2^)
**PCDTBT:PCBM**	−10.66	0.77	0.48	3.94	11.5	401
**PCDTBT:PCBM:(TTP)GaCl 1:0.05**	−11.14	0.78	0.53	4.61	8.2	489
**P3HT:PCBM**	−12.09	0.60	0.50	3.63	5.3	289
**P3HT:PCBM:(TTP)GaCl 1:0.05**	−12.90	0.61	0.54	4.25	3.7	318

## Data Availability

The data that support the findings of this study can become available by the corresponding authors upon reasonable request.

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
