# Peer review of "Efficient and Stable Air-Processed Ternary Organic Solar Cells Incorporating Gallium-Porphyrin as an Electron Cascade Material"

_nanomaterials, 2023, doi:10.3390/nano13202800_

Round 1
Reviewer 1 Report
The manuscript investigates the effect of a tetraphenyl GaCl porphyrin and an octaethylporphyrin GaCl porphyrin to be used as electron cascade in ternary organic bulk heterojunction films. The tetraphenyl GaCl porphyrin with low concentration was shown to improve PCE. Photostability was also shwon to improve for the best performed devices.
The paper is well written and the text is clear with logical scientific flow. The conclusions are consistent with the evidence and arguments presented. I believe the topic will be of great interest to the photovoltaics community. Thus, I recommend the manuscript to be published in Nanomaterials.
Minor editing of English language required
Author Response
Dear Referee,
We would like to sincerely thank you all for your time and efforts to review our manuscript and provide fruitful instructions. The manuscript was carefully edited for mistakes.

Reviewer 2 Report
Vasilopoulou et al. incorporate Gallium-Porphyrin into the active layer of organic solar cells. The device performance was achieved and the mechanisms were discussed. This work is recommended to be accepted after the following issues are addressed.
1. In Figure 2e, the CBs of (TPP)GaCl (-3.3) and (OEP)GaCl (-3.2) seem at the same level, better adjust it to consistent with the real value.
2. In Figure 3, the absorption spectra don’t show characterized absorption peak of PCDTBT, which is different with the reported results, please check.
3. In Figure 4a-c, the images are too dark, better adjust the color scale to make it clearer.
4. In Figure 7a, the content of (TPP)GaCl seems equal to PCBM, but the real content is very small, better revise it to avoid misunderstanding.
5. In the introduction, better cite some recent review papers to facilitate the readere to know the recent advances about OSCs, such as: Materials Futures, 2023, 2(3): 032102.
6. Some discussions should be added about the improvement in FF and PCE, an in-depth analysis can be found in J. Semicond., 2021, 42(9): 090501.
English Language is OK.
Author Response
Dear Referee,
We would like to sincerely thank you all for your time and efforts to review our manuscript and provide fruitful instructions. We carefully revised our manuscript based on these instructions. We hope that the revised version meets your expectation and it is now suitable for publication to Nanomaterials. Our detailed response to your comments is listed below.
REFEREE REPORT:
Referee: 2
Vasilopoulou et al. incorporate Gallium-Porphyrin into the active layer of organic solar cells. The device performance was achieved and the mechanisms were discussed. This work is recommended to be accepted after the following issues are addressed.
- In Figure 2e, the CBs of (TPP)GaCl (-3.3) and (OEP)GaCl (-3.2) seem at the same level, better adjust it to consistent with the real value.
Response: We thank the referee for the comment. Figure 2e has been revised according to referee suggestion.
- In Figure 3, the absorption spectra don’t show characterized absorption peak of PCDTBT, which is different with the reported results, please check.
Response: We sincerely thank the Referee for the comment. Figures 3a and b have been revised. The absorption at around 570nm corresponds to the PCDTBT.
- In Figure 4a-c, the images are too dark, better adjust the color scale to make it clearer.
Response: We thank the referee from the comment. Figure 4 has been revised according to referee suggestion.
- In Figure 7a, the content of (TPP)GaCl seems equal to PCBM, but the real content is very small, better revise it to avoid misunderstanding.
Response: We sincerely thank the referee for the comment. Figure 7a has been revised according to the referee comment.
- In the introduction, better cite some recent review papers to facilitate the readere to know the recent advances about OSCs, such as: Materials Futures, 2023, 2(3): 032102.
Response: We thank the referee for the comment. The manuscript has been revised according to the proposed comment.
- Some discussions should be added about the improvement in FF and PCE, an in-depth analysis can be found in J. Semicond., 2021, 42(9): 090501.
Response: We sincerely thank you for the comment. Further discussion for the improvement in FF and PCE is added to the revised manuscript.

Reviewer 3 Report
The paper by Soultati et al. reports the use of GaCl porphyrins as a third component to C60 fullerene-based organic solar cells. The authors demonstrate an improved performance and stability upon the addition of the porphyrins in small amounts to the host PCDTBT:PCBM and P3HT:PCBM bulk-heterojunction blends, and support their findings using spectroscopic and morphological measurements. Although the study of ternary systems may be of some interest in the community at the moment, the objects of study, fullerene-based blends of PCDTBT and P3HT, are quite outdated. The paper is in general scientifically sound, however the mechanisms and reasons for the improved performance and stability have not been thoroughly and clearly accentuated. In addition, the following comments must be addressed before considering the paper for publication in Nanomaterials:
1) The first sentence in the Abstract refers to the porphyrin molecules being synthesized. However, the paper does not report any details of the synthesis.
2) The authors several times refer to the ternary blends obtained using the studied porphyrins as 'an efficient cascade system'. However, the paper does not specify and provide evidence for the mechanism of charge (electron and hole) transfer between the species within this cascade. The provided energy level diagram (Fig. 2) does not necessarily explain which charge transfer reactions take place within the ternary blends upon photoexcitation.
3) Reference 46 reports a semitransparent perovskite, not an organic solar cell.
4) In Line 135, it is written that 'in the (OEP)GaCl porphyrin’s structure, each of the pyrrole rings is substituted by ethyl groups'. However, according to Figure 1b for (OEP)GaCl the pyrrole rings are not substituted by ethyl groups, but rather the ethyl groups are attached to pyrrole rings. This should be clarified.
5) In Line 174, it is mentioned that 'Figure 2e reveals the perfect matching of both gallium porphyrins energy levels with that (those) of polymer donors'. This claim should be explained in more detail. How is the perfect matching of energy levels defined in this case? How are charges transferred within such a cascade?
6) In Fig. 3a, the authors should explain why the Soret band corresponding to (TPP)GaCl in Fig. 3a has a lower intensity for the higher amount of (TPP)GaCl (1:0.1) than for its lower amount (1:0.05).
7) In Figs. 3c and 3d, what is the origin of the intense band at around 2300 cm-1?
8) In Line 225, it is written: 'Upon introducing (TPP)GaCl to the P3HT blend, the PL intensity decreases, indicating an enhanced electron injection rate from P3HT to (TPP)GaCl and efficient suppression of electron-hole recombination'. How is the possible P3HT PL quenching by the porphyrin related to the suppressed free charge recombination?
9) In Line 298, it is written 'It is shown that the GaCl-porphyrins acknowledged that even in cases where the donor and acceptor materials demonstrate an ideal electronic relationship in a bulk heterojunction, it is essential for them to exhibit stability under typical environmental conditions to ensure an extended lifetime of the device.' The authors should rewrite this sentence as it is not entirely clear what is meant. Also, where is this shown?
10) In Line 349, it is written 'It is noted that this PCE enhancement is independent from the ternary film thickness, since the insertion of porphyrin of low concentration in the binary blend slightly change it (~145 nm).' This sentence is unclear and should be rephrased. Do you mean that the device thickness remained unchanged after adding the porphyrins?
11) In Line 357, it is written 'The JSC and FF enhancement suggests reduced recombination losses...' However, a stronger evidence suggesting suppressed recombination and/or linking recombination with JSC and FF is necessary. J-V improvements could also come from small variations in absorption, device thickness, etc. Also, which type of recombination is considered in this case: geminate or non-geminate? bimolecular, trap-induced or other?
12) In Line 391, it is written 'Moreover, at high VEFF values (VEFF > 1), JPH,SAT of the ternary OSCs is higher than that of the binary ones, resulted in enhanced exciton generation rate...'. How does a higher photogenerated current density result in enhanced exciton generation rate? A clarification and/or rephrasing is necessary.
Finally, the authors should clarify whether the improvement in the device performance is due to the improved energetics resulting in an energetic cascade or other factors, such as improved nanomorphology.
The quality of English across the text should be checked and improved.
1) Check singular/plural endings. For example, in Page 3, Line 100: 'region of the visible spectrum' should be replaced with 'regions of the visible spectrum'. In Page 4, Line 110: 'by mean of optical spectroscopy' should be replaced with 'by means of optical spectroscopy', etc.
2) Check other grammatical constructs. For example, in Page 4, Line 121: 'which assigned' should be replaced with 'which is assigned', etc.
3) Check typos carefully. For example, in Page 9, Line 200: 'FTIR transmittamce' should be replaced with 'FTIR transmittance', etc.
4) The quality of written English should be improved. For example, in Page 18, Line 395: 'Except the device efficiency...' could be replaced with 'In addition to the device performance...'.
Author Response
Dear Referee,
We would like to sincerely thank you all for your time and efforts to review our manuscript and provide fruitful instructions. We carefully revised our manuscript based on these instructions. We hope that the revised version meets your expectation and it is now suitable for publication to Nanomaterials. Our detailed response to your comments is listed below.
REFEREE REPORT:
Referee: 3
1) The first sentence in the Abstract refers to the porphyrin molecules being synthesized. However, the paper does not report any details of the synthesis.
Response: We sincerely thank the referee for the comment. Details about the synthesis of the GaCl porphyrinn materials were added to the revised manuscript.
2) The authors several times refer to the ternary blends obtained using the studied porphyrins as 'an efficient cascade system'. However, the paper does not specify and provide evidence for the mechanism of charge (electron and hole) transfer between the species within this cascade. The provided energy level diagram (Fig. 2) does not necessarily explain which charge transfer reactions take place within the ternary blends upon photoexcitation.
Response: We thank you for the comment. We re-perform cyclic voltammetry and measure CV along with ferrocene. The estimation of HOMO and LUMO level is highlighted in the revised manuscript. The HOMO level of (TPP)GaCl and (OEP)GaCl is approximately -5.6 eV and -5.5 eV, respectively, while LUMO is estimated at -3.8 eV for the (TPP)GaCl and -3.9 eV for the (OEP)GaCl. It is clearly seen that the energy levels of the porphyrin compounds match with the energy levels of donor materials (PCDTBT and P3HT) and the PCBM-acceptor, acting as an efficient electron-cascade material.
3) Reference 46 reports a semitransparent perovskite, not an organic solar cell.
Response: We apologize for the mistake. The reference has been changed.
4) In Line 135, it is written that 'in the (OEP)GaCl porphyrin’s structure, each of the pyrrole rings is substituted by ethyl groups'. However, according to Figure 1b for (OEP)GaCl the pyrrole rings are not substituted by ethyl groups, but rather the ethyl groups are attached to pyrrole rings. This should be clarified.
Response: We thank the referee for the comment. In OEP(GaCl) the ethyl groups are attached to pyrrole rings. We apologize for the mistake.
5) In Line 174, it is mentioned that 'Figure 2e reveals the perfect matching of both gallium porphyrins energy levels with that (those) of polymer donors'. This claim should be explained in more detail. How is the perfect matching of energy levels defined in this case? How are charges transferred within such a cascade?
Response: We thank the referee for the comment. As already mentioned we re-measure CV, re-estimate the HOMO and LUMO levels of porphyrin compounds and re-make energy level diagram (comment 2). In addition, EQE spectra shown in the revised manuscript supports the fact that porphyrins do not contribute to light harvesting but only in charge transfer.
6) In Fig. 3a, the authors should explain why the Soret band corresponding to (TPP)GaCl in Fig. 3a has a lower intensity for the higher amount of (TPP)GaCl (1:0.1) than for its lower amount (1:0.05).
Response: We agree with the referee comment. We carefully checked the spectroscopic measurements and found out the mistake in diagram 3a. The correct spectrum of (TPP)GaCl 1:0.1 is that of (TPP)GaCl 1:0.05 and reverse. We apologize for the mistake. The spectra have been changed.
7) In Figs. 3c and 3d, what is the origin of the intense band at around 2300 cm-1?
Response: We thank you for the comment. The peak is related to the background CO2 in the spectrometer.
8) In Line 225, it is written: 'Upon introducing (TPP)GaCl to the P3HT blend, the PL intensity decreases, indicating an enhanced electron injection rate from P3HT to (TPP)GaCl and efficient suppression of electron-hole recombination'. How is the possible P3HT PL quenching by the porphyrin related to the suppressed free charge recombination?
Response: We thank the referee for the comment. The sentence has been rephrased.
9) In Line 298, it is written 'It is shown that the GaCl-porphyrins acknowledged that even in cases where the donor and acceptor materials demonstrate an ideal electronic relationship in a bulk heterojunction, it is essential for them to exhibit stability under typical environmental conditions to ensure an extended lifetime of the device.' The authors should rewrite this sentence as it is not entirely clear what is meant. Also, where is this shown?
Response: We sincerely thank the referee for the comment. Changes have been made and are highlighted in the revised manuscript.
10) In Line 349, it is written 'It is noted that this PCE enhancement is independent from the ternary film thickness, since the insertion of porphyrin of low concentration in the binary blend slightly change it (~145 nm).' This sentence is unclear and should be rephrased. Do you mean that the device thickness remained unchanged after adding the porphyrins?
Response: We thank you for the comment. The thickness of the photoactive didn’t change with the insertion of the porphyrins due to the very low concentration of them in the ternary film. We have rewritten the sentence.
11) In Line 357, it is written 'The JSC and FF enhancement suggests reduced recombination losses...' However, a stronger evidence suggesting suppressed recombination and/or linking recombination with JSC and FF is necessary. J-V improvements could also come from small variations in absorption, device thickness, etc. Also, which type of recombination is considered in this case: geminate or non-geminate? bimolecular, trap-induced or other?
Response: We thank the referee for the comment. Since the thickness of the film remained unchanged and according to electrical characterization and the EQE spectra of the fabricated OSCs we believe that the improved JSC and FF indicates reduced recombination losses. In addition, a shift observed in the PL spectra shown in Figure 3e and indicates that trap-assisted mechanism is dominated in the ternary film. More details have been added in the revised manuscript.
12) In Line 391, it is written 'Moreover, at high VEFF values (VEFF > 1), JPH,SAT of the ternary OSCs is higher than that of the binary ones, resulted in enhanced exciton generation rate...'. How does a higher photogenerated current density result in enhanced exciton generation rate? A clarification and/or rephrasing is necessary.
Response: We sincerely thank the referee for the comment. The equation GMAX = JPH,SAT/qL was used to estimate exciton generation rate (GMAX), where q I the elementary charge and L is the thickness of the photoactive layer. As the thickness of the active layer did not change the GMAX is proportional to the JPH,SAT.

Author Response
Dear Referee,
We would like to sincerely thank you all for your time and efforts to review our manuscript and provide fruitful instructions. We carefully revised our manuscript based on these instructions. We hope that the revised version meets your expectation and it is now suitable for publication to Nanomaterials. Our detailed response to your comments is listed below.
REFEREE REPORT:
Referee: 4
- Authors need to confirm the energy levels much better method than CV presented in the paper. Because the peaks in the CV are not clear, as they did not show any notable oxidation and reduction peaks. Authors should remeasure the CV of the compounds along with ferrocene or use other methods like PESA or UPS to determine the energy levels.
Response: We sincerely thank the referee for the comment. We re-perform cyclic voltammetry and measure CV along with ferrocene. The estimation of HOMO and LUMO level is highlighted in the revised manuscript.
- EQE spectra of the optimized PV devices need to be included.
Response: We thank you for the comment. EQE spectra have been added to the revised manuscript.
- The efficiency enhancement upon 3rd component was too less, did authors tried increasing doping? Please display the results
Response: We thank the referee for the comment. Upon increasing the concentration for 0.05%v/v to 0.1%v/v, the performance of the ternary OSCs was decreased as shown in Figure S8 in the Supporting Information, which can be attributed to the rougher surface morphology and the suppressed crystallinity of the ternary active layers (Figure 4 and S2). The formation of large aggregates in the ternary films were observed, when porphyrin compounds with high concentration were inserted in the binary photoactive films, which hindered exciton generation and charge separation, thus leading to lower PCE values.
- The location of the third component is very important in determining the process involved in the ternary blend OPV systems. Thus, it is necessary to provide some evidence or explanation for it. Authors might refer to the following related works (Sol. RRL 2017, 1700158; ACS Appl. Mater. Interfaces 2018, 10, 16, 13748–13756).
Response: We sincerely thank the referee for the comment. We perform contact angle measurements of neat materials, binary and ternary films in order to investigate the location of the porphyrins in the ternary blends. We also estimate the wetting coefficient of the third component (ie porphyrins) and we found out that porphyrin as third component is localized at the donor/acceptor interface. The results are highlighted in the revised manuscript.
- AFM image (a-c) looks strange, please double check or repeat the experiment to get better image.
Response: We thank the referee from the comment. Figure 4 has been revised according to referee suggestion.
- Important reference should be cited in the manuscript introduction and wherever it is needed.
(https://doi.org/10.1002/cssc.202101345; https://doi.org/10.1002/aenm.201700782; https://doi.org/10.1038/s43246-023-00395-y )
Response: We thank the referee for the comment. The manuscript has been revised according to referee suggestion.

Round 2
Reviewer 2 Report
The authors have made satisfying revisions. This work is recommended to be accepted at present form.
Reviewer 3 Report
Thanks for the provided responses.
Reviewer 4 Report
Since authors did revise the manuscript according to the comments raised, thus I believe that work can be considered for a publication.
Since authors did revise the manuscript according to the comments raised, thus I believe that work can be considered for a publication.